# Impact of Delta SARS-CoV-2 Infection on Glucose Metabolism: Insights on Host Metabolism and Virus Crosstalk in a Feline Model

**DOI:** 10.3390/v16020295

**Published:** 2024-02-15

**Authors:** Matthew T. Rochowski, Kaushalya Jayathilake, John-Michael Balcerak, Miruthula Tamil Selvan, Sachithra Gunasekara, Craig Miller, Jennifer M. Rudd, Véronique A. Lacombe

**Affiliations:** 1Department of Physiological Sciences, College of Veterinary Medicine, Oklahoma State University, Stillwater, OK 74078, USA; mrochow@okstate.edu (M.T.R.);; 2Harold Hamm Diabetes Center, Oklahoma City, OK 73104, USA; 3Department of Veterinary Pathobiology, College of Veterinary Medicine, Oklahoma State University, Stillwater, OK 74078, USA; miruthula.tamil_selvan@okstate.edu (M.T.S.); sachithra.gunasekara@okstate.edu (S.G.); craigmillerdvm@gmail.com (C.M.); jennifer.rudd@okstate.edu (J.M.R.)

**Keywords:** COVID-19, insulin, cortisol, angiotensin 2, GLUT, AMPK

## Abstract

Severe Acute Respiratory Syndrome Coronavirus 2 (SARS-CoV-2) causes enhanced mortality in people with metabolic and cardiovascular diseases. Other highly infectious RNA viruses have demonstrated dependence on glucose transport and utilization, so we hypothesized that SARS-CoV-2 infection could lead to alterations in cellular and whole-body glucose metabolism. Twenty-four healthy domestic cats were intratracheally inoculated with B.1.617.2 (delta) SARS-CoV-2 and samples were collected at 4- and 12-days post-inoculation (dpi). Blood glucose and circulating cortisol concentrations were elevated at 4 and 12 dpi. Serum insulin concentration was statistically significantly decreased, while angiotensin 2 concentration was elevated at 12 dpi. SARS-CoV-2 RNA was detected in the pancreas and skeletal muscle at low levels; however, no change in the number of insulin-producing cells or proinflammatory cytokines was observed in the pancreas of infected cats through 12 dpi. SARS-CoV-2 infection statistically significantly increased GLUT protein expression in both the heart and lungs, correlating with increased AMPK expression. In brief, SARS-CoV-2 increased blood glucose concentration and cardio-pulmonary GLUT expression through an AMPK-dependent mechanism, without affecting the pancreas, suggesting that SARS-CoV-2 induces the reprogramming of host glucose metabolism. A better understanding of host cell metabolism and virus crosstalk could lead to the discovery of novel metabolic therapeutic targets for patients affected by COVID-19.

## 1. Introduction

The delta variant emerged during the COVID pandemic in late 2020 resulting in more hospitalizations and deaths than any previous variant [1]. Still, over 3 years following the rapid emergence of this virus, SARS-CoV-2 continues to circulate and garner attention. Historical context with coronaviruses, knowledge of their increased mutation rates as RNA viruses, and their known ability for interspecies transmission, underscore the distinct possibility that a future variant could re-emerge at pandemic levels. In order to be better prepared for potential future pandemics, it is paramount to continue analyzing pandemic viral strains and to determine the mechanisms that enhance viral fitness compared to previous variants. SARS-CoV-2 is particularly devastating for patients with comorbidities, including metabolic diseases such as cardiovascular disease (CVD) and diabetes [2]. Elevated blood glucose appears to be favorable for viral replication, and SARS-CoV-2 infection can lead to elevated blood glucose [3], although the underlying mechanisms are not fully elucidated. Although COVID increases the risk for development of CVD [4], the pathophysiological mechanism underlying COVID-19-induced CVD remains to be fully elucidated. Potential underlying pathophysiological mechanisms include thrombosis, inflammation, and alterations in the renin–angiotensin–aldosterone system (RAAS) [5,6]. Since the underlying mechanisms for CVD post-COVID-19 are multi-variable, further defining how SARS-CoV-2 infection modulates cardiac glucose metabolism will contribute to our understanding of how COVID-19 may induce CVD.

Pancreatic beta cell destruction leads to the development of type 1 diabetes due to reduced insulin secretion. Subsequently, insulin-stimulated glucose transporter (GLUT) trafficking to the cell surface, the rate-limiting step in glucose uptake, becomes impaired. GLUT4 is the primary insulin-sensitive isoform that becomes downregulated during diabetes, while GLUT1 is the primary basal isoform. Both of these GLUTs are responsible for the uptake of glucose into the cell to allow for glycolysis and can be regulated by AMPK, a master metabolic switch [7]. While SARS-CoV-2 is a respiratory infection, it can cause detrimental effects to other organs, including the pancreas. Evidence has been mounting that SARS-CoV-2 infection could lead to the development of new-onset diabetes [3,4]. Currently, theories on the mechanisms include (1) an auto-immune attack of the pancreas, which has been observed with other viruses; (2) inflammatory damage in the pancreas due to cytokine storm or (3) by direct infection of beta-cells [8,9,10]. Although it has been shown that isolated beta cells are susceptible to SARS-CoV-2 infection [11], confirmation of direct viral infection and destruction of insulin-producing beta cells within the pancreas of infected individuals has yet to occur.

To our knowledge, there have been few mechanistic studies investigating metabolic alterations in the lung and insulin-sensitive tissue (e.g., striated muscle) following SARS-CoV-2 infection. In this study, we hypothesized that SARS-CoV-2 infection leads to alterations in whole-body and cellular glucose metabolism of the host. A better understanding of how these alterations in glucose metabolism affect virus–host interaction could identify novel metabolic therapeutic targets for patients affected by COVID-19.

## 2. Materials and Methods

### 2.1. Virus

SARS-CoV-2 virus isolate B.6.617.2 (delta) was obtained from BEI Resources and passaged up to 6 times in Vero cells (CCL-81, ATCC, Manassas, VA, USA) in Dulbecco’s Modified Eagle Medium with 5% fetal bovine serum and antibiotics. The TCID50 of viral stock solution was calculated using the Reed and Muench method [12], as previously described [13].

### 2.2. Study Animals

All experiments were approved by the Animal Ethics Committee of Oklahoma State University (IACUC 20-48 STW). Details regarding study animals have been previously published [13]. Briefly, thirty 1-year-old (control *n* = 6, 4 dpi *n* = 12, and 12 dpi *n* = 12) and sex-matched (15 male, 15 female) specific-pathogen-free cats were obtained from Marshall BioResources (North Rose, NY, USA). Twenty-four cats were infected intratracheally with SARS-CoV-2 (B.6.617.2 delta variant) and sacrificed 4- and 12-days post-infection (dpi, *n* = 12/group). An additional 6 cats were sham-inoculated (controls). Cats to be inoculated with SARS-CoV-2 were individually housed within Animal Biosafety Level 3 (ABSL-3) animal rooms at Oklahoma State University and given access to dry/wet food and water ad libitum. Animals used for sham inoculation were group-housed in an AAALAC internationally accredited animal facility at Oklahoma State University. Animals were allowed 30 days for acclimation prior to initiation of the study. Baseline weights, body temperatures, and clinical evaluation were collected prior to inoculation.

### 2.3. Virus Challenge

Details regarding the B.6.617.2 delta variant infection protocols have been previously published [13]. Briefly, cats were anesthetized with ketamine (4 mg/kg) (Covetrus, Dublin, OH, USA), dexmedetomidine (20 µg/kg) (Orion, Espoo, Finland), and butorphanol (0.4 mg/kg) (Zoetis, Gilrona, Spain) intramuscularly. Cats were inoculated at the level of the distal trachea using 1 mL of Dulbecco’s Modified Eagle Medium (DMEM) (GibcAqo, Carlsbad, CA, USA), containing 1.26 × 10^6^ TCID_50_ of SARS-CoV-2 isolate B.6.617.2 (delta), as previously described [13]. Sham-inoculated cats were administered 1 mL of inoculation media. At day 4 and day 12 post-infection, cats infected with delta SARS-CoV-2 (*n* = 12 per time-point) and sham-infected control cats (*n* = 3 per timepoint) were anesthetized, and then humanely euthanized (pentobarbital > 80 mg/kg) for tissue collection. Following euthanasia, necropsy tissues including cranial lung, left ventricle, gastrocnemius, and pancreas were collected. Tissues were divided and either frozen at −80 °C or collected into tissue cassettes and fixed in 10% neutral-buffered formalin for 5 days prior to transferring to 70% ethanol.

### 2.4. H&E Staining and Immunofluorescence

Briefly, 5 µm thick paraffin sections of pancreatic tissues were collected onto positively charged slides prior to staining with hematoxylin and eosin (H&E) for microscopic evaluation or immunofluorescence assay (IFA), as previously described [14]. For IFA, slides were rehydrated using previously outlined methods of toluene and ethanol changes [14] followed by antigen retrieval using Citrate Unmasking Solution (Cell Signal, Danvers, MA, USA). Slides were blocked with 10% goat serum for 1 h followed by overnight incubation with a rabbit polyclonal anti-insulin antibody (4590, Cell Signaling Technology, Danvers, MA, USA). Anti-rabbit Alexa Fluor 555 was used as the secondary antibody, as well as DAPI for a nuclear counterstain (Cell Signaling Technology, Danvers, MA, USA). All slides were assigned a quantitative histologic score based on relative total number of cells and fluorescent intensity by a board-certified veterinary pathologist blinded to study groups to ensure scientific rigor and reproducibility.

### 2.5. Droplet Digital Polymerase Chain Reaction Analyses (ddPCR)

Copies of SARS-CoV-2 viral RNA and feline angiotensin-converting enzyme 2 (fACE2) RNA were quantified in tissues by droplet digital PCR (ddPCR), as previously described [13,14]. Briefly, RNA was isolated using 50–80 mg of tissue was homogenized in Trizol reagent (Invitrogen, Waltham, MA, USA). RNA was quantified via absorbance (A_260_) using Gen5 software with Biotek synergy HT hardware on a take3 plate (BioTek, Winooski, VT, USA). RNA was treated with DNAse (Ambion AM1906M), then converted to complementary DNA (cDNA) using the High-Capacity cDNA Reverse Transcription Kit and random primers (Applied Biosystems, Foster City, CA, USA). All ddPCR analyses for SARS-CoV-2 and fACE2 RNA were performed using previously published primers and probes [13,14] (Appendix A). PCR reaction master mixtures were 5 µL One-Step RT-ddPCR Advanced Kit for Probes Supermix (no dUTPs) (Bio-Rad, Hercules, CA, USA); 2 µL reverse transcriptase; 1 µL 300 mM Dithiothreitol (DTT); 1 µL triplex probe assay for N1, N2, and RPP30 detection (Bio-Rad, Hercules, CA, USA); 2 µL RNase-free water; and 9 µL cDNA. Samples of cDNA were run in duplicate, adding 20 µL of each sample toQX200 droplet generator (Bio-Rad, Hercules, CA, USA) and transferred to 96-well plates and sealed. Data were analyzed using QuantaSoft™ Software (Bio-Rad, Hercules, CA, USA) and expressed as Log10 (copies/mL).

### 2.6. Blood Metabolites

Blood samples were collected prior to infection (baseline) and at 4 and 12 dpi. Glucose and ketone colorimetric assays were performed according to the manufacturer’s instructions (Cayman Chemical Company, Ann Arbor, MI, USA product #’s 10009582 and 700190, respectively). Feline cortisol and insulin ELISA assays were performed according to the manufacturer’s instructions (Cayman Chemical Company, Ann Arbor, MI, USA product #’s 500360 Mercodia, Sweden product # 10-1233-01, respectively). Angiotensin 2 ELISA assay was performed according to manufacturer’s instructions (Enzo Life Sciences, Farmingdale, NY, USA product # ADI-900-204).

### 2.7. Western Blotting

Western blots were performed as previously described [15]. Briefly, tissues were homogenized in 1:500 dilution of radioimmunoprecipitation assay (RIPA) buffer (ThermoFisher, product #89901, Waltham, MA, USA) with protease inhibitor cocktail (Sigma Aldrich, product #P8340, St. Louis, MO, USA), and quantified via bicinchoninic assay according to manufacturer’s instructions (ThermoFisher, product #23227, Waltham, MA, USA). Equal amounts of protein (30–70 μg) were loaded and transferred to a polyvinylidine fluoride membrane (BioRad, Hercules, CA, USA). Membranes were washed, then incubated in primary antibodies overnight with constant agitation at 4 °C. Primary antibodies were chosen based on their 100% sequence homology with the protein of interest and previously validated by our laboratory [15] (GLUT1, 1:500 Abcam product #ab115730; GLUT4, 1:750 Abcam product #ab33780; Phospho-AMPKα Thr-172, 1:500 Cell Signaling product #2535; AMPKα 1:500 Cell Signaling product #2532). The next day, membranes were washed and incubated with secondary antibody. Antibody-bound proteins were quantified using chemiluminescence reaction (Super Signal Max Sensitivity Substrate, product #34095, ThermoFisher, Waltham, MA, USA) and autoradiography. Membranes were stripped (ThermoFisher, product #21063, Waltham, MA, USA) and re-probed using β-Actin or Calsequestrin (β-actin, 1:300 Santa Cruz product #sc-47778, Calsequestrin, 1:500 ThermoFisher product #PA1-913). Band density was quantified using GelPro Analyzer (Media Cybernetics, Rockville, MD, USA), and data were represented as arbitrary relative units normalized to controls, as previously described [15].

### 2.8. Statistical Analysis

Normality and homogeneity of data were determined using Shapiro–Wilk and Levene’s tests, respectively. Differences in means were determined using 1-way ANOVA or Student’s *t*-test. Data are reported as mean ± standard error and statistical significance defined as *p* < 0.05.

## 3. Results

### 3.1. Feline SARS-CoV-2 Infection Induced Elevated Blood Glucose and Angiotensin 2 Concentrations, as well as Decreased Insulin Concentration

Infection kinetics and immunopathogenesis of this feline model were previously reported by our group [13]. Briefly, cats displayed lethargy, increased respiratory effort, and wheezing, which peaked at 4–5 dpi. In the present study, we reported that cats infected with the delta variant of SARS-CoV-2 had elevated blood glucose concentration at 4 dpi (*p* = 0.001), which returned to baseline at 12 dpi (Figure 1A). In contrast, ketone concentration was not statistically significantly changed following SARS-CoV-2 infection (*p* = 0.11 Ctr vs. 4 dpi, Figure 1B). Interestingly, we noticed a decreased serum insulin concentration along with increased angiotensin 2 concentration at 12 dpi (Figure 1C *p* = 0.01, Figure 1D *p* = 0.03). As expected, circulation cortisol was statistically significantly increased in response to SARS-CoV-2 infection (*p* < 0.05) (Figure 1E).

#### 3.1.1. Feline SARS-CoV-2 Infection Induced Low-Level Viremia and Viral RNA Detection in Peripheral Tissues

We previously reported significant levels of SARS-CoV-2 RNA in the lungs and cardiac muscle of cats [13]. Here, we reported that delta SARS-CoV-2 RNA was also detectable in skeletal muscle (*p* = 0.0477) and pancreas (*p* = 0.13) at 4 dpi (Figure 2A,C). No statistically significant change in ACE2 expression was observed in any of these tissues following SARS-CoV-2 infection (Figure 2B,D). A very low level of viremia was detected via TCID_50_ (e.g., 3.16 × 10^2^ TCID_50_/mL) at 4 dpi in infected feline plasma.

#### 3.1.2. SARS-CoV-2 Infection Did Not Induce Pancreatic Beta Cell Destruction or Inflammatory Cytokine Production

Since we detected SARS-CoV-2 RNA in the pancreas at 4 dpi and observed decreased insulin secretion at 12 dpi, we investigated whether SARS-CoV-2 infection could lead to the destruction of pancreatic beta cells in our feline model. To this end, histopathology and immunofluorescence assays were performed on the pancreas using anti-insulin antibodies to identify beta cells of the pancreas. Following imaging and blind scoring of both the number of cells within the beta islets (Figure 3A) and fluorescent intensity (Figure 3B), we did not observe a decreased number of beta cells or reduced insulin production. Histological changes were also not observed in the pancreas of infected cats as compared with sham-inoculated controls, with the exception of one cat that exhibited mild lymphoplasmacytic inflammation affecting one pancreatic islet at 12 dpi (Appendix A). We also quantified pro-inflammatory cytokines in the pancreas using ddPCR; however, we did not observe any increase in IL-1β, IL-6, IL-8, or TNFα RNA levels at 4 or 12 dpi (Figure 4A–D).

#### 3.1.3. SARS-CoV-2 Infection Induced Increased GLUT Protein Expression in the Cardio-Respiratory System but Not in Peripheral Insulin-Sensitive Tissues

We measured protein expression of basal and insulin-sensitive glucose transporters (GLUT1 and GLUT4, respectively) in total lysates of the lung and heart using Western blotting. GLUT1 protein expression in the lung was increased approximately two-fold throughout SARS-CoV-2 infection (*p* = 0.03 and 0.04, Figure 5A). Pulmonary GLUT4 protein expression was also increased at 4 dpi in infected cats by approximately 1.5-fold (*p* = 0.049, Figure 5B). In the heart, both total GLUT1 and GLUT4 protein expression were increased at 4 dpi (*p* = 0.004 and 0.011 by 2.25- and 6.5-fold, respectively). Cardiac GLUT4 protein content was also statistically significantly upregulated at 12 dpi (*p* = 0.0005, Figure 5D), with a similar statistical trend for GLUT1 protein content (*p* = 0.09, Figure 5C). We further determined whether an increase in glucose transport following SARS-CoV-2 infection was specific to infected tissue, namely, the cardiorespiratory system, or whether it could also affect peripheral insulin-sensitive tissues such as skeletal muscle. We noticed a statistical trend toward an increase in GLUT1 protein expression at 12 dpi (*p* = 0.07), but no change in total GLUT4 protein expression in skeletal muscle following SARS-CoV-2 infection (Figure 5E,F).

#### 3.1.4. AMPK Activation in the Heart and Lung during SARS-CoV-2 Infection

Since AMP-activated protein kinase (AMPK) is a master metabolic regulator, we measured total AMPK (tAMPK) protein content and activated/phosphorylated AMPK (pAMPK) using Western blotting. Following infection, pAMPK (at the active site Thr172) /tAMPK remained unchanged in all of tissues (Figure 6A–C). In the lung, both pAMPK and tAMPK protein expression (compared to their respective loading controls) showed an increasing statistical trend at 4 dpi (*p* = 0.06 and 0.07, respectively), and were both significantly increased at 12 dpi (Figure 6A). In the heart, pAMPK and tAMPK protein expression was significantly upregulated by ~2-fold at 4 dpi (*p* = 0.03 and 0.004, respectively). We also observed an increase in phosphorylated AMPK and a statistical trend of greater tAMPK protein expression in the heart at 12 dpi (*p* = 0.034 and *p* = 0.051, respectively) (Figure 6B). In contrast, there was no change in AMPK expression in the skeletal muscle of infected vs. healthy cats (Figure 6C).

## 4. Discussion

While a multitude of human tissues naturally express ACE2, the entry receptor for SARS-CoV-2, finding a suitable animal model has been challenging. Rodents are commonly used for translational research but must be genetically modified to express human ACE2. However, even genetically modified mice only express the human ACE2 receptor in their epithelial tissues [16], which may limit our ability to analyze whole-body alterations in the renin–angiotensin–aldosterone system (RAAS). We previously demonstrated that the domestic cat is a useful comparative model for SARS-CoV-2 infection since it naturally expresses similar ACE2 isoform as humans [17]. In addition, the domestic cat is a good model for diabetes as it can spontaneously develop both type 1 and type 2 diabetes [18]. In terms of pancreas structure and cell-type quantity, the domestic cat is also very similar to humans [19,20]. For instance, both cats and humans have been found to develop amyloidosis in the pancreas during development of type 2 diabetes [21,22].

Cats infected with the delta variant SARS-CoV-2 generally demonstrated similar clinical signs to cats during wild-type-strain infection [13], which included lethargy, increased respiratory effort, and wheezing. Similar to findings reported in human patients [9] with COVID-19, we observed transient hyperglycemia in this feline model, which could reflect a transient state of diabetes [23] and/or an increased metabolic demand in immune cells following SARS-CoV-2 infection [24]. This is germane to the fact that there is growing evidence that the metabolic pathways (e.g., glycolysis, glutaminolysis, and oxidative metabolism) primarily regulate the functions of immune cells [25]. Infected cats also displayed increased angiotensin 2 concentration, which may have induced the observed decrease in circulating insulin at 12 dpi. Future studies could determine whether similar levels of elevated angiotensin 2 could decrease insulin production and induce diabetes.

Delta SARS-CoV-2 RNA was previously been reported in the lung and cardiac muscle of cats infected with the delta strain without significant alterations in ACE2 mRNA expression [13]. In this study, we detected SARS-CoV-2 viral RNA in the pancreas and skeletal muscle of ~ half of the infected cats at 4 dpi, but at lower levels than previously reported in the lung and cardiac muscle of this feline model (10^4^–10^7^ copies/mL). While we cannot rule out potential active replication of the virus in these tissues, these lower levels of viral RNA may be explained by the presence of the virus within circulating blood, which was supported by detectable low levels of the virus in the blood via TCID_50_.

Due to the expression of ACE2 receptors on the pancreas, pancreatic injury may occur as a result of SARS-CoV-2 infection, which could lead to infection-induced type 1 diabetes [8,11,26]. Although studies showed that pancreatic islet cells are susceptible to infection in vitro, confirmation has been lacking in vivo [27,28]. Thus, we further aimed to determine if SARS-CoV-2 induces destruction of the beta islets, which could lead to the development of insulin-dependent diabetes. We did not detect a reduction in beta cell numbers through 12 dpi, nor decreased insulin production from the islets of infected cats. In addition, we did not observe any elevation in cytokines associated with both an immune response and the development of diabetes [29] (IL-1β, IL-6, IL-8, and TNFα) using ddPCR. Although viral RNA was detected at 4 dpi, no changes in pro-inflammatory cytokines were observed through 12 dpi. Taken together, our data suggest that it is unlikely that SARS-CoV-2 induces the destruction of pancreatic beta islets, either via direct infection or indirectly through an inflammatory response, in our feline model. While it seems that human enteroviruses are most likely to induce diabetes, the mechanism with the most supported mechanism is molecular mimicry [30], which has also been shown to occur with SARS-CoV-2 proteins [31,32,33]. Therefore, future studies may be helpful in determining the long-term effects of SARS-CoV-2 infection on pancreas function.

One molecule that may be able to link SARS-CoV-2 with both diabetes and CVD is angiotensin 2 (ang2) [34,35]. There is continued speculation regarding the dynamics and function of the ACE2 protein activity during SARS-CoV-2 infection [3,36,37,38]. It has been previously demonstrated that SARS-CoV-2 infection leads to decreased cell-surface ACE2 due to endocytosis with the virus [36,39]. This would therefore decrease the amount of functional ACE2, potentially leading to the accumulation of ang2. Furthermore, it has been shown that accumulation of ang2 could also reduce insulin secretion, lead to insulin resistance, and impair glucose metabolism [40,41,42,43]. Here, we did not observe any change in ACE2 RNA expression in the pancreas or skeletal muscle in SARS-CoV-2 infected cats, yet we observed an accumulation of ang2 and a reduction in insulin in these cats at 12 dpi. These data support the possibility that the SARS-CoV-2 inhibition of ACE2 may lead to a pre-diabetic state in susceptible individuals. More long-term studies are needed to identify the underlying pathophysiological mechanisms.

To determine the alterations in glucose uptake following SARS-CoV-2 infection, we measured protein expression of basal and insulin-sensitive GLUT in the lung, heart, and skeletal muscle. GLUT-1 and -4 protein expression was increased in the lungs during the acute phase of infection. It has been demonstrated that viruses may be capable of modulating host cell metabolism in a way that favors necessary ATP synthesis to support viral replication or to allosterically activate glycolysis to further support the energy demands of viral replication [44,45]. There is also evidence that SARS-CoV-2 and influenza virus produce similar pathophysiological responses and depend on similar replication mechanisms in vivo [46,47]. For instance, these RNA viruses produce a more severe infection and maintain a higher mortality rate in diabetic and obese patients [48,49]. In addition, our laboratory demonstrated that influenza increased glucose concentration in bronchoalveolar lavage and pulmonary GLUT4 trafficking in healthy and diabetic mice [50], and our data here suggest that pulmonary glucose transport was enhanced following SARS-CoV-2 infection.

SARS-CoV-2 infection also leads to greater GLUT protein expression in the heart. Therefore, one could speculate that increased cardiac glucose uptake enhances glycolysis to compensate for the energy demand of the replicating virus. Similarly, increased glucose uptake and metabolism have been reported in the decompensating heart as opposed to the typically preferred fatty acid oxidation pathway [51]. While this shift in metabolic substrate is currently thought to be protective [52], it is possible that shifting toward glucose metabolism could have unknown downstream consequences. Furthermore, we observed an upregulation of GLUT1 in the skeletal muscle, which may represent a potential compensatory mechanism. Since skeletal muscle is the largest glucose sink in the body, an increase in the expression of basal GLUTs has been reported in response to hyperglycemia [53,54].

We next wanted to explore possible pathways that could allow SARS-CoV-2 to enhance glucose transport. Therefore, we measured AMPK activation (by Western blotting pAMPK levels). AMPK is a key metabolic sensor in the cell, which facilitates energy production following lowering the ATP/ADP ratio, and AMPK activation is the target of the most used diabetic drug (e.g., metformin), as it increases GLUT trafficking to the cell surface and reduces blood glucose concentrations [55]. As expected, pAMPK was increased at 12 dpi in the lung, with a statistical trend toward an increase at 4 dpi. Similarly, we noticed an increase in cardiac AMPK activation at both 4 and 12 dpi. This was correlated with an increase in GLUT expression in the lung and the heart. Although AMPK was activated in infected tissues (i.e., heart and lung), it remained unchanged in the skeletal muscle, suggesting a possible activation of AMPK in direct response to SARS-CoV-2 infection. Activation of AMPK has been observed during various other viral infections [56,57]. However, to our knowledge, this is the first report of AMPK activation during SARS-CoV-2 infection in the cardiorespiratory system. Future studies using a greater sample size should be performed to determine the contributions of AMPK in the activation of glucose transport as well as the inflammatory response in SARS-CoV-2 infected cells. Both of these proteins are central metabolic targets that could be modulated during SARS-CoV-2 infection to alter viral success and host survival.

## 5. Conclusions

This study identified novel pathophysiological mechanisms underlying SARS-CoV-2 infection-induced hyperglycemia, using a translational feline model that parallels human COVID-19 [58]. We demonstrated that SARS-CoV-2 infection induces alterations of key glucose metabolism proteins (i.e., GLUT and AMPK) in the cardio-respiratory system, without inducing a significant inflammatory response of the pancreatic beta islets in our feline model. These data suggest that SARS-CoV-2 infection induces metabolic changes that then could enhance viral replication. Thus, insight from this study sheds some light on the host cell metabolism and virus crosstalk, which could lead to the development of complementary metabolic therapeutic strategies for patients with COVID-19 infection.

## Figures and Tables

**Figure 1 viruses-16-00295-f001:**
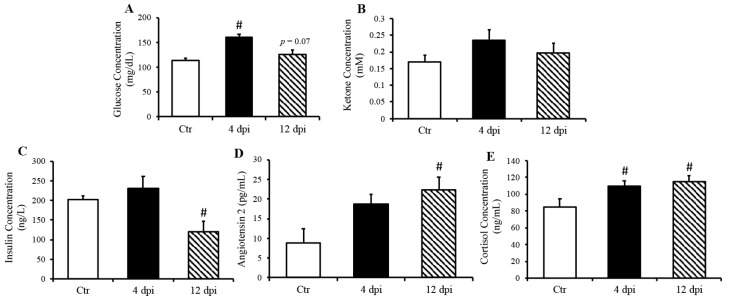
Increased fasted serum glucose and cortisol concentrations in cats, 4- and 21-days post-infection (dpi), along with increased angiotensin 2 and decreased insulin 12 dpi, but no change in ketone concentration. Mean ± SE of fasted serum glucose (**A**) ketone (**B**), insulin (**C**), angiotensin 2 (**D**), and cortisol (**E**) concentrations in control and infected cats; *n* = 6–12/group; # *p* < 0.05 vs. non-infected control (Ctr).

**Figure 2 viruses-16-00295-f002:**
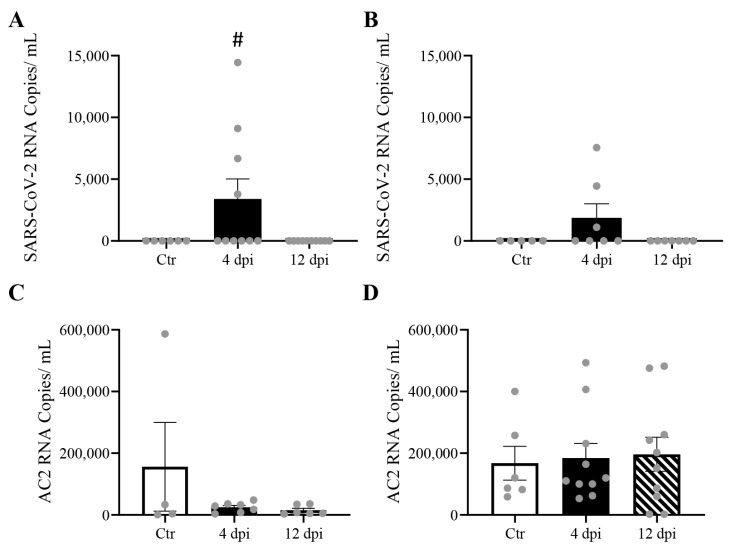
Mild SARS-CoV-2 viremia was detected, leading to viral RNA detection in both the pancreas and skeletal muscle at 4 days post-infection (dpi), without change in angiotensin-converting enzyme 2 (ACE2) RNA expression. Mean ± SE copies/mL of SARS-CoV-2 (**A**,**B**), and ACE2 mRNA(**C**,**D**) of skeletal muscle and pancreas of control vs. 4 and 12 dpi cats; *n* = 4–12/group; # *p* < 0.05 vs. Ctr.

**Figure 3 viruses-16-00295-f003:**
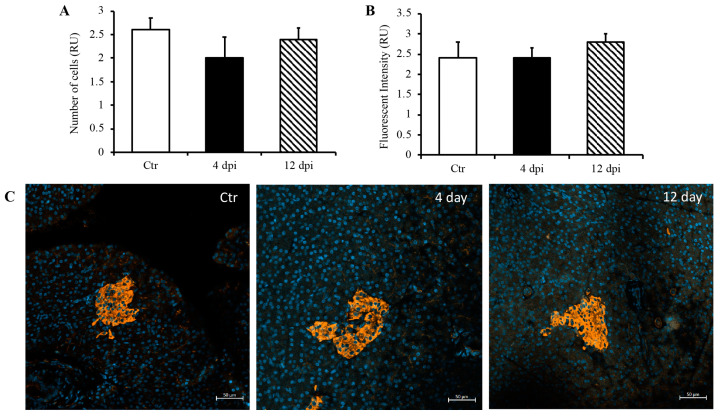
No destruction of beta islets or decreased insulin production through 12 dpi. Blindly scored relative units (RU) of the number of beta cells (**A**) and fluorescent intensity (**B**) in a given field of the pancreas. (**C**) Representative immunofluorescence images of pancreas using DAPI and anti-insulin antibodies. *n* = 6–12/group.

**Figure 4 viruses-16-00295-f004:**
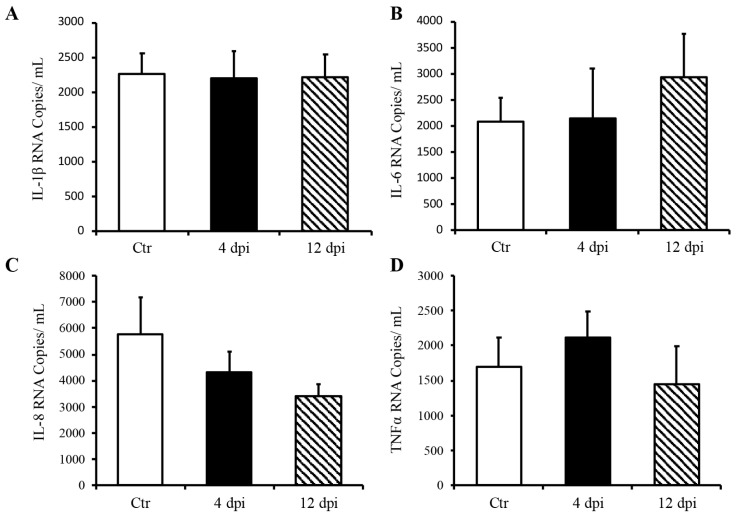
Delta SARS-CoV-2 infection did not induce a pro-inflammatory response in the pancreas of these cats through 12 dpi. Mean ± SE of relative IL-1b (**A**), IL-6 (**B**), IL-8 (**C**), and TNFα (**D**) RNA in the pancreas; *n* = 6–12/group; Ctr = non-infected, dpi = days post-infection.

**Figure 5 viruses-16-00295-f005:**
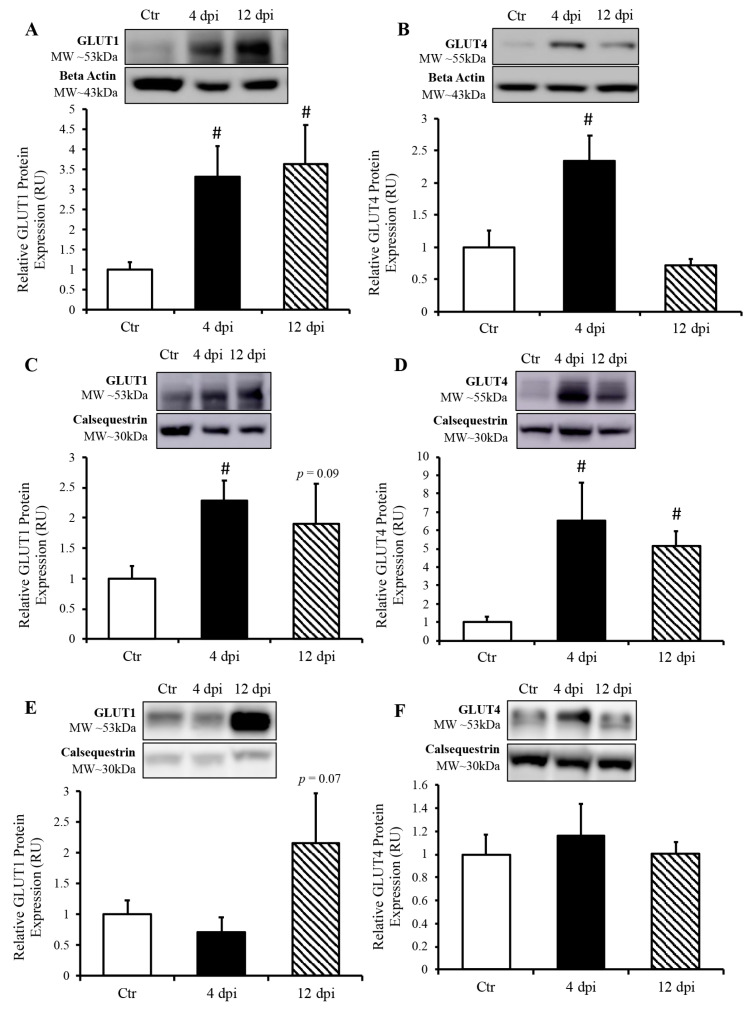
SARS-CoV-2 infection induced increased protein expression of glucose transporters (GLUT) in the lung and heart, but not in the skeletal muscle. (**Top panels**): representative Western blot from total lysate; beta-actin or calsequestrin were used as a loading control; representative bands were obtained from the same membrane. (**Bottom panels**): Mean ± SE of relative total protein content of basal GLUT1 and insulin-sensitive GLUT4 in the lung (**A**,**B**), respectively, heart (**C**,**D**), respectively and skeletal muscle (**E**,**F**), respectively of control (Ctr) and infected cats; n = 6–12/group; # *p* < 0.05 vs. Ctr.

**Figure 6 viruses-16-00295-f006:**
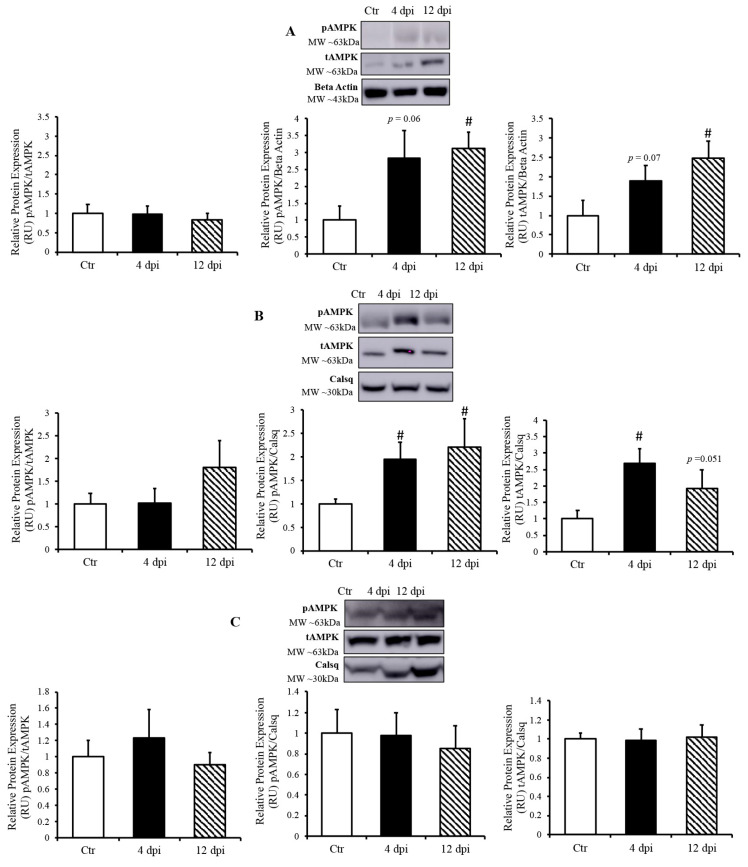
AMP-activated kinase (AMPK) activated only in SARS-CoV-2 infected tissues. (**Top panels**)**:** representative Western blot from total lysate; beta-actin or calsequestrin was used as a loading control; representative bands were obtained from the same membrane. (**Bottom panels**): Mean ± SE of phosphorylated AMPK/ total AMPK, phosphorylated AMPK/loading control, and total AMPK/loading control in the lung (**A**), heart (**B**), and skeletal muscle (**C**) of control (Ctr) and infected cats, *n* = 6–12/group; # *p* < 0.05 vs. Ctr.

## Data Availability

Data will be made available upon reasonable request.

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
