# Peer review of "Impact of Delta SARS-CoV-2 Infection on Glucose Metabolism: Insights on Host Metabolism and Virus Crosstalk in a Feline Model"

_viruses, 2024, doi:10.3390/v16020295_

Round 1

Reviewer 1 Report

Comments and Suggestions for Authors

In this study, a feline model is utilized to test the hypothesis that SARS-CoV-2 infection could induce alterations in cellular and/or whole-body glucose metabolism.  This is important since it has been shown that elevated glucose levels promote viral replication.  In this model, domestic cats were infected intratracheally with the Delta variant of SARS-CoV-2 and, at 4 and 12 days p.i., the effects on blood glucose, insulin concentration and angiotensin 2 concentration were determined.  Indeed, both blood glucose and angiotensin 2 concentration were elevated and insulin concentration was decreased.  Further, SARS-CoV-2 RNA was detectable at very low levels in the pancreas (albeit in only 3/6 of infected animals).  Despite these findings, there were no detectable changes in the number of insulin-producing cells, the amounts of proinflammatory cytokines or ACE2 expression in the pancreas.  Indeed, there is yet to be direct confirmation of viral infection and destruction of beta cells in infected individuals.  However, there were detectable increases in the expression of glucose transporters GLUT1 AND GLUT4 in heart and lung tissue, consistent with virus-induced enhancement of pulmonary glucose transport., correlating with increased AMPKJ expression.  Based on these findings, the authors speculate that, although the virus does not appear to directly affect the pancreas, it could modulate glucose metabolism through an AMPK-dependent mechanism, which, in turn, enhances viral replication.

This manuscript does not support any firm conclusions about the effect of SARS-CoV-2 infection on glucose metabolism.  Rather, the data are more suggestive in nature.  Therefore, it is considered to have only a moderate impact on the field, but does identify further avenues of investigation that could be impactful.

Specific points

1)    Lines 177-8:  It is not accurate to say that ketone concentration was unchanged following infection.

2)    Line  :  It is not accurate to conclude that ACE2 RNA expression is unchanged in the infected pancreas.

3)    Lines 235: The increase in GLUT1 expression appears to be much more than two-fold.

4)    Lines 239-40: Aren’t both GLUT1 and GLUT4 upregulated at 12 hrs?

Comments on the Quality of English Language

Minor improvement needed.

Author Response

The authors thank the referees for the thorough and constructive review of the manuscript. We addressed all the reviewers’ comments, and as a result, we believe that the current version of the manuscript is improved. Revised changes to the manuscript are in red.

Specific points

  • Lines 177-8: It is not accurate to say that ketone concentration was unchanged following infection.

We reformulated the sentence “was not statistically significantly change”, as suggested. Please refer to line 182 of the revised manuscript.

  • Line 194: It is not accurate to conclude that ACE2 RNA expression is unchanged in the infected pancreas.

This paragraph was reworded. Please refer to lines 196-200 of the revised manuscript.

  • Lines 235: The increase in GLUT1 expression appears to be much more than two-fold.

A different representative blot was used to better represent the data as ~ a two-fold difference. Please refer to revised Fig 5, Panel C

  • Lines 239-40: Aren’t both GLUT1 and GLUT4 upregulated at 12hrs?

We thank the reviewer for this insightful point. Cardiac GLUT4 protein content was also statistically significantly upregulated 12 dpi (p=0.0005, Fig 5C, D), with a similar statistical trend for GLUT1 protein content (p=0.09). The sentence was revised accordingly. Please refer to lines 240-242 of the revised manuscript.

Reviewer 2 Report

Comments and Suggestions for Authors

Rochowski et al explore the influence of the SARS-CoV-2 delta variant on alterations to glucose metabolism using a feline model. They find SARS-CoV-2 infection related hyperglycemia is likely not caused by inflammation or destruction of pancreatic tissue but alterations in glucose uptake due to increased glucose transporter expression. This work provides insight into how SARS-CoV-2 viral infection can have such negative effects on individuals with metabolic comorbidities. Though the results are sound, the manuscript can be hard to follow at times and should be revised for clarity.

Major comments:

Materials and methods lines 79-90 information about the animals used in this study is insufficient. Could the authors please include information on animal age, sex, and general health of animals at the beginning of the study

Section 3.1.1 and the figure 2 legend are very confusing as written, can the authors please clarify language used and add additional information on viral titers and references to figure 2

Line 128 Could the authors please provide a table with primers and probes used in this study

Minor comments:

Lines 34-35 “mutations rates” should be “mutation rates”

Line 44 “…cardiovascular disease (CVD)” should just be CVD, this was defined already in line 41

Lines 86-87 is confusing as worded. Could the authors please clarify wording and include if animals were allowed food/water ad libitum

Line 174 The authors should consider adding a clinical summary and viral titer of individual animals in the study

Line 175 reported should be present tense

Lines 191-198 is confusing as written. Could the authors please introduce figure 2 more clearly. As written it is unclear which parts of the figure refer to which tissues/RNA targets.

Lines 195-198 are confusing as written. Could the authors please clarify this sentence and include information on the viral titers found in the tissues mentioned as well as plasma

Comments on the Quality of English Language

Minor issues with language outlined in comments but okay overall.

Author Response

The authors thank the referees for the thorough and constructive review of the manuscript. We addressed all the reviewers’ comments, and as a result, we believe that the current version of the manuscript is improved. Revised changes to the manuscript are in red.

Major comments:

Materials and methods lines 79-90 information about the animals used in this study is insufficient. Could the authors please include information on animal age, sex, and general health of animals at the beginning of the study:

Clarification was added: thirty (control n=6, 4 dpi n=12, and 12 dpi n=12) age- (1-year-old) and sex-matched (15 male, 15 female). Please refer to lines 83-87 of the revised manuscript.

Section 3.1.1 and the figure 2 legend are very confusing as written, can the authors please clarify language used and add additional information on viral titers and references to figure 2, Lines 191-198 is confusing as written. Could the authors please introduce figure 2 more clearly. As written it is unclear which parts of the figure refer to which tissues/RNA targets.

This section was modified to enhance clarify, as suggested. We previously reported significant levels of SARS-CoV-2 RNA in the lungs and cardiac muscle of cats [13]. Here, we reported that delta SARS-CoV-2 RNA was also detectable in skeletal muscle as well as the pancreas, without any statistical changes of ACE mRNA expression between tissues. Information on plasma viral titers was also added. Please refer to lines 197-202 of the revised manuscript and revised Fig 2 legend.

Line 128 Could the authors please provide a table with primers and probes used in this study:

Primers used were added to Table 1S.

Minor comments:

Lines 34-35 “mutations rates” should be “mutation rates” corrected, thank you

Line 44 “…cardiovascular disease (CVD)” should just be CVD, this was defined already in line 41 corrected, thank you

Lines 86-87 is confusing as worded. Could the authors please clarify wording and include if animals were allowed food/water ad libitum: Specification was added line 90.

Line 174 The authors should consider adding a clinical summary and viral titer of individual animals in the study: This was previously reported, and a brief summary was added lines 178-179

Line 175 reported should be present tense: “Data was reported” was corrected to “data is reported” (line 173).

Lines 195-198 are confusing as written. Could the authors please clarify this sentence and include information on the viral titers found in the tissues mentioned as well as plasma:

This sentence was modified and the viral titers was included, as suggested. Please refer to lines 197-201 of the revised manuscript.

Reviewer 3 Report

Comments and Suggestions for Authors

                Using a feline model of SARS-CoV2 infection that was established a few years ago, the authors report a descriptive analysis of glucose metabolism associated factors, viral infections of muscle and pancreas, proinflammatory cytokine RNA levels, GLUT protein expression and AMPK expression.  Overall the study design is straightforward and the data presented will be interesting to the field.  However as noted below, several of the data in the study appear to be inconsistent with the conclusions that are drawn or require additional experimentation to provide clear conclusions.  Thus the manuscript as currently presented may be premature to report at this time as it appears to represent the result of one infection study without any apparent follow up to address aspects of the data that were unclear.

Major Points:

1.        Fig 2:  Fig. 2C:  I’m surprised that no SARS-CoV-2 was found in the pancreas at 12 dpi. Given the large error bar at 4 dpi, was the detection of SARS-CoV-2 at 4 hours significant?

2.       Fig. 2D:  The sustained ~ 10X decrease in ACE2expression in the pancreas looks to be highly significant, yet the authors conclude that there is no change in ACE2 expression in the pancreas during infection.  Thus the data appear to be inconsistent with the conclusion that is drawn.

3.       Fig. 2:  what do panels E and F represent?  They are not discussed in the legend.

4.       Fig. 4C appears to show a pretty convincing 2X decrease in IL8 RNA at 12 dpi post infection.  Was this observation significant?  This should be explicitly discussed in the text.

5.       The authors refer to statistical trends (e.g. tAMPK protein expression in the heart (Fig. 6);  GLUT 1 expression in Fig. 5).  These are potentially interesting observations and, optimally, the experiment should be repeated, to fully assess the significance of these trends to increase the impact of the manuscript. 

Minor Points:

1.        Fig.1B:  While the authors conclude from these data that the ketone concentration is unchanged following SARS-CoV2 infection, the data are clearly trending in the same fashion as the glucose concentration in Fig. 1A.  I’m curious what the p values are for these data in the study.

2.       There are some minor typos throughout the manuscript. For example, in line 181, SARS-CoV2 should be SARS-CoV-2.  On line 227 in the legend to Fig. 4 panel D is not denoted.  There are others as well.  I would suggest that the authors carefully review the manuscript to correct these minor presentation flaws.

3.       Line 195:  The authors state that the viral RNA was significantly lower compared to their previously reported data.  To assist the reader, I would recommend that they explicitly report the copies per ml in each study and provide the statistics to support this claim.

4.       Fig 2:  The heading on the legend refers to viremia – yet no data are presented regarding virus in the blood.

Comments on the Quality of English Language

1.       There are some minor typos throughout the manuscript. For example, in line 181, SARS-CoV2 should be SARS-CoV-2.  On line 227 in the legend to Fig. 4 panel D is not denoted.  There are others as well.  I would suggest that the authors carefully review the manuscript to correct these minor presentation flaws.

Author Response

The authors thank the referees for the thorough and constructive review of the manuscript. We addressed all the reviewers’ comments, and as a result, we believe that the current version of the manuscript is improved. Revised changes to the manuscript are in red.

Major Points:

  1. Fig 2:  Fig. 2C:  I’m surprised that no SARS-CoV-2 was found in the pancreas at 12 dpi. Given the large error bar at 4 dpi, was the detection of SARS-CoV-2 at 4 hours significant?

SARS-CoV-2 was detected in skeletal muscle of 4 cats and in the pancreas of 3 cats. Please refer to revised fig 2 which included individual data points These changes were only statistically significant in the pancreas. The p-values were added in lines 199 of the revised manuscript.

  1. Fig. 2D:  The sustained ~ 10X decrease in ACE2expression in the pancreas looks to be highly significant, yet the authors conclude that there is no change in ACE2 expression in the pancreas during infection. Thus, the data appear to be inconsistent with the conclusion that is drawn.

Because of the variability of the data (especially in the control group), the data was not statistically significant. Please refer to revised fig 2 which included individual data points.

  1. Fig. 2:  what do panels E and F represent?  They are not discussed in the legend:

Thank you for bringing this oversight to our attention, which has been corrected. Since previous mRNA expression of SARS-CoV-2 and ACE2 was reported in the lungs and cardiac muscle of our feline model (Ref#13), we only reported results in skeletal muscle and pancreas. Please refer to revised fig 2.

  1. Fig. 4C appears to show a pretty convincing 2X decrease in IL8 RNA at 12 dpi post infection. Was this observation significant? This should be explicitly discussed in the text.

The p value for IL8 RNA using one-way ANOVA was 0.145.

  1. The authors refer to statistical trends (e.g. tAMPK protein expression in the heart (Fig. 6); GLUT 1 expression in Fig. 5). These are potentially interesting observations and, optimally, the experiment should be repeated, to fully assess the significance of these trends to increase the impact of the manuscript. 

We thank the reviewer for bringing up this point and we agree that additional experiments will be required to confirm our findings. This point has been added in the revised discussion (please refer to line 383-385).

Minor Points:

  1. 1B: While the authors conclude from these data that the ketone concentration is unchanged following SARS-CoV2 infection, the data are clearly trending in the same fashion as the glucose concentration in Fig. 1A. I’m curious what the p values are for these data in the study.

Ctr vs 4 dpi: 1-tail t-test: P=0.113, 2-tail t-test: P=0.226. This information is added line 184 of the revised manuscript.

  1. There are some minor typos throughout the manuscript. For example, in line 181, SARS-CoV2 should be SARS-CoV-2.  On line 227 in the legend to Fig. 4 panel D is not denoted.  There are others as well.  I would suggest that the authors carefully review the manuscript to correct these minor presentation flaws.

These minor typos were corrected, thank you.

  1. Line 195:  The authors state that the viral RNA was significantly lower compared to their previously reported data. To assist the reader, I would recommend that they explicitly report the copies per ml in each study and provide the statistics to support this claim.

This statement was moved to the discussion and reference ranges were added, please see lines 309-311.

  1. Fig 2:  The heading on the legend refers to viremia – yet no data are presented regarding virus in the blood.

Data on viremia is added lines 201-202 of the revised manuscript.

Round 2

Reviewer 3 Report

Comments and Suggestions for Authors

                The authors have responded adequately to the points raised in the original round of peer review.  Given the lack of statistical significance that remains associated with some interesting trends in the data, the study could arguably still be premature to report at this time (as was pointed out in the previous critique).